# Supermarket Circulars Promoting the Sales of ‘Healthy’ Foods: Analysis Based on Degree of Processing

**DOI:** 10.3390/nu12092877

**Published:** 2020-09-21

**Authors:** Alyne Michelle Botelho, Anice Milbratz de Camargo, Kharla Janinny Medeiros, Gabriella Beatriz Irmão, Moira Dean, Giovanna Medeiros Rataichesck Fiates

**Affiliations:** 1Graduate Program in Nutrition, Nutrition in Foodservice Research Centre, Federal University of Santa Catarina, University Campus João David Ferreira Lima-Trindade, Florianópolis, SC 88040-900, Brazil; alyne.botelho@posgrad.ufsc.br (A.M.B.); anice.camargo@posgrad.ufsc.br (A.M.d.C.); kharla.medeiros@ufsc.br (K.J.M.); gabriella.irmao@grad.ufsc.br (G.B.I.); 2Institute for Global Food Security, School of Biological Sciences, Queen’s University Belfast, Belfast BT9 5DL, UK; moira.dean@qub.ac.uk

**Keywords:** supermarket circulars, ultra-processed, food label, health claims

## Abstract

The health and wellness food sector grew 98% from 2009 to 2014 in Brazil, the world’s fourth-biggest market. The trend has reached supermarket circulars, which recently started to feature whole sections advertising health and wellness-enhancing foods. This study identified food items advertised in circulars’ specific sections of two Brazilian supermarket chains (one regional, one national) during a 10-week period. Foods were classified according to degree of food processing and presence/type of claims on their front-of-pack (FoP) labels. Comparison between groups of Unprocessed/Minimally Processed foods vs. Ultra-processed foods and presence/type of claims employed Pearson chi-square test. From the 434 alleged health and wellness-enhancing foods advertised, around half (51.4%) were classified as Ultra-processed. Presence of reduced and increased nutrient-content claims was significantly higher in labels of Ultra-processed foods. Most frequent claims addressed sugar and fibre content. Brazilian supermarket circulars were found to be promoting the sale of Ultra-processed foods in their health and wellness sections, leading to a situation that can mislead the consumer and bring negative health outcomes.

## 1. Introduction

Healthy eating is essential for health promotion and protection, and as a determinant factor in preventing chronic non-transmissible diseases [1]. Nevertheless, the access to a healthy and adequate diet is proving to be a challenge for modern societies, with the eating practices of Brazilians in different stages of life and across all socioeconomic strata being far from what is considered desirable [2]. Consumption of processed and ultra-processed foods has been growing exponentially in the Brazilian population and is considered a contributing factor for the increased prevalence of obesity and non-communicable diseases in the country [3,4]. Consequently, a new version of the Dietary Guidelines for the Brazilian Population was published, instructing individuals to limit the consumption of processed foods, avoid consumption of ultra-processed foods, and choose fresh and minimally processed foods as the core of their diets. According to the Guidelines, a healthy diet is based on the consumption of natural or minimally processed foods; and of dishes and meals containing such foods [1].

Published in 2015, the Dietary Guidelines for the Brazilian Population established specific eating directives based on degree of food processing, employing a classification system later improved and published under the name of NOVA (a name, not an acronym) [5]. According to NOVA, industrial formulations containing little or no fresh ingredients and food additives to add colour, flavour, texture, and additional sensory properties to unprocessed foods and preparations containing them are classified as Ultra-processed foods [1,5]. The nature of the processes and ingredients used in their manufacture, and their displacement of unprocessed or minimally processed foods and freshly prepared dishes and meals, make ultra-processed foods intrinsically unhealthy. In spite of this, as foods typically energy-dense, rich in sugar, fat, and salt, they are hyper-palatable and cheap, which contributes to their high consumption. Ultra-processed foods are poor in dietary fibre, protein, vitamins, and minerals; and additives contained in their formulation increase shelf-life without increasing their cost [1,5,6].

On the other hand, dietary patterns based on dishes and meals made from a variety of unprocessed or minimally processed plant foods, prepared, seasoned, and cooked with processed culinary ingredients and complemented with processed foods are the healthier ones [6]. Examples of ultra-processed foods include but are not limited to cookies, fizzy drinks, confectionery items, cereal bars, bottled sauces, instant noodles, and sweetened milk-based beverages. Their intake is discouraged, while that of unprocessed or minimally processed foods is encouraged to constitute the core items of the population’s diet [1]. Unprocessed (or natural) foods are edible parts of plants (seeds, fruits, leaves, stems, roots) or of animals (muscle, offal, eggs, milk), and also fungi, algae and water, after separation from nature [5]. Minimally processed foods, that together with unprocessed foods make up NOVA group 1 are unprocessed foods altered by industrial processes such as removal of inedible or unwanted parts, drying, crushing, grinding, fractionating, roasting, boiling, pasteurisation, refrigeration, freezing, placing in containers, vacuum packaging or non-alcoholic fermentation. None of these processes add salt, sugar, oils or fats, or other food substances to the original food. Their main aim is to extend the life of grains (cereals), legumes (pulses), vegetables, fruits, nuts, milk, meat and other foods, enabling their storage for longer use, and often to make their preparation easier or more diverse [6].

Despite of what is recommended by official guidelines, people have complex and diversified interpretations about the concept of healthy eating, which reflect their personal, social, cultural, and environmental experiences [7]. The concept of healthy eating is frequently unclear for individuals and is not understood and interpreted identically by all [8]. This can lead to the adoption of different practices in the name of healthy eating [9].

A definitive and universally accepted concept of healthy eating does not exist, but its association with better health and disease prevention is largely recognised [10]. During the second half of the last century, the increased availability and diversity of (un)healthy foods considerably modified the concept of what constitutes a healthy diet [11]. As the focus changed from a nutrient-based approach to a food-based one, food classification systems based on degree of processing were proposed [12]. In this context, the higher the processing degree to which a food has been submitted, the lower is the frequency in which it should be ingested as part of a healthy diet [5].

As ultra-processed foods tend to be energy-dense and low-cost, low energy cost could be one mechanism linking ultra-processed foods with high consumption and consequent negative health outcomes [13]. In Brazil however, the total cost of diets based on natural or minimally processed foods is still lower than the cost of diets based on ultra-processed foods. Relatively expensive perishable foods such as some vegetables, fruits, and fish are and should be consumed with other natural or minimally processed foods that have lower prices, such as rice, beans, potatoes, cassava, and other staple traditional Brazilian foods. Calculations based on Brazilian household budget surveys show that diets based on fresh and minimally processed foods, and dishes and meals made with these foods and culinary ingredients, are cheaper than diets made of ultra-processed foods, as well as being healthier [1].

According to Euromonitor International, Brazilian population’s interest in healthy foods has increased between 2009 and 2014 [14]. The health and wellness food sector accounts for a US $35 billion market each year and is expected to grow on average 5% per year until 2021. The ‘free from’ food category presents the largest growth, stimulated by the increased (but unrelated to dietetic intolerance) consumption of gluten and lactose-free foods [14].

Nutrient-content claims are regulated in Brazil as ‘Complementary Nutrition Information’ (CNI), defined as ‘representations which affirm, suggest or imply that a product has particular nutritional properties especially, but not solely restricted to its energy, protein, fat, carbohydrate and fibre content, and also vitamin and mineral content’. CNIs may refer to absolute or relative/comparative nutrient content of food products using terms as: ‘without’, ‘no’, ‘absence’, ‘low content’, ‘does not contain’; and ‘presence’, ‘contains’, ‘high content’, ‘rich’, ‘source of’. Regulation on parameters for the voluntary display of CNI on front-of-pack (FOP) labels of packaged products exist since 2012 [15], but claims are allowed without consideration for foods’ whole nutrient composition or degree of processing. Therefore, an ultra-processed food product containing high levels of sugar and/or sodium may display a nutrient-content claim of ‘low fat’ or ‘vitamin rich’ on its label.

Health claims are regulated by a resolution published in 1999 and amended in 2004, which determines that a ‘health property claim’ is one that affirms, suggests or implies a relationship between the food/ingredient and diseases or health-related conditions [16]. It may also describe a physiological role which assists normal growth, development and functions, and contributes to health maintenance and reduced risk of diet-related disease [16].

Nutrition labelling is designed to help consumers make healthier food choices, provided they understand the vocabulary or layout used to display nutritional information. For this reason, the highlight of positive characteristics in food products by means of nutrient or health claims is regarded as a marketing strategy to promote sales [17], as many are found on unhealthy food items [18]. Claims have the potential to both inform and mislead consumers, depending on the information that is highlighted and the kind of product displaying this information [19]. The highlight of positive characteristics as nutrient claims on front-of-pack labels can generate a ‘health halo’ effect, when consumers’ assessment of a single positive characteristic of the food affects their judgment about the quality of the food as a whole [20]. Health halos can be conferred by claims concerning just one nutrient, because consumers often make generalisations about the overall health of a product based on one piece of information found on labels [21].

Notably, the ‘free-from’ consumption trend has influenced the content of supermarket circulars. Together with images of products’ front-of-pack labels, circulars present products’ prices to aid consumers to plan their shopping, but also significantly influencing their shopping decisions. National and regional supermarket chains have started to dedicate whole sections of circulars to the promotion of foods designated by them (possibly together with manufacturers), as health and wellness-enhancing [14].

‘Wellness’ refers to the positive, subjective state that is opposite to illness [22], an evolving process toward achieving one’s full potential [23]. Wellness is positive/affirming and holistic, and encompasses lifestyle, spiritual, and environment wellbeing domains; it also accounts for the physical, mental, and social domains implied in health, and thus health is dependent on sufficient wellness [23]. As consumers today are more health conscious than ever before, and the food and beverage industry is driven by consumer demand and popular trends, the health and wellness trend is increasingly prevalent. Health and wellness foods such as energy bars, gluten and dairy-free products, products containing organic/prebiotic ingredients, and fortified/functional foods are significantly more expensive than the regular offering, which makes them an expensive luxury in many emerging markets [14].

Research on the content of supermarket circulars found that most advertised foods are unhealthy and not conducive to the adoption of a diet in line with official recommendations [24,25,26,27,28,29,30,31]. Studies reported that circular offers did not contribute towards an environment that supports healthy eating behaviour, but most only assessed the products advertised on the front pages and not in the entire circulars. Additionally, they were mostly conducted in European countries, North America, and Australia. Only one study was identified reporting the analysis of Latin American (Brazilian) supermarkets’ entire circulars [31].

The situation can mislead consumers and negatively impact their health, as supermarket circulars are reportedly used by consumers as planning tools [32,33], and can predict subsequent memory of the advertised product or brand [34]. Foods advertised in circulars are also further promoted in-store and positioned in strategic places such as end-of-aisle or islands located in places of great circulation, in order to significantly influence shopping decisions [35].

The present study seeks to extend this stream of research focusing on circulars not only advertising novel products or promotions, but dedicating sections specifically to the promotion of as health and wellness-enhancing foods.

To our knowledge, no papers about the quality of products advertised in health and wellness-enhancing sections of supermarket circulars have been published. The aim of this study was to analyse, according to degree of food processing, the quality of foods advertised in the health and wellness-enhancing sections of supermarket circulars from Brazilian supermarkets, and identify presence and type claims on the front-of-pack labels of such products.

## 2. Materials and Methods

This cross-sectional study was undertaken in the capital city of Santa Catarina state (Florianópolis), southern Brazil. The capital was chosen out of convenience (near the university where the research team works), and also because it is the state’s second largest city. The supermarket chains for circulars’ collection were defined according to the frequency of circular distribution (fortnightly) and the presence of a specific section advertising health and wellness-enhancing foods. Two supermarket chains (one national and one regional) with stores in the capital distributed circulars with the aforementioned characteristics. Both pertain to the group of 50 companies (national chain: 14th position; regional chain: 44th position) with the highest gross sales (R $2,711,219,166.00 and R $928,708,550.00, national and regional chain, respectively) in 2018 according to the ranking of the Brazilian Association of Supermarkets [36]. The national chain has a total of 29 stores in Brazil, of which six are located in Florianópolis; the regional chain has a total of 22 stores in the state and nine are located in Florianópolis.

To collect the circulars, one store from each chain was conveniently chosen, both located in a residential middle-class neighbourhood near two university campuses. A total of 20 circulars (10 from each chain) were collected in situ or downloaded from the supermarket website, between October 2018 and April 2019, at 15-day intervals. Nineteen printed circulars were retrieved; one circular from the regional chain was only available online. Collection was paused between December and February to avoid the influence of seasonal offers in circulars (Christmas and New Years’ holiday season and summer vacation).

All images of products advertised in the health and wellness sections from retrieved circulars were analysed. Different package sizes and shapes of the same product from the same brand were counted as one (e.g., spaghetti and penne pasta). Different flavours of the same food item (e.g., grape juice and orange juice from the same brand) were counted as different items. The same happened to similar products by different brands.

Manufacturers’ websites and supermarket stores were then visited to retrieve the ingredient lists of all products (except for unprocessed foods items) in order to proceed with the categorisation according to degree of processing. Foods were categorised into one of four groups as (a) unprocessed and minimally processed foods (U/MP); (b) processed culinary ingredients (PCI); (c) processed foods (P); and (d) ultra-processed foods (UP) (Appendix A) [5,6]. A decision flowchart specifically developed to guide the categorisation was used [37]. Whenever the flowchart was not applicable, a conservative criterion [38] was applied (i.e., product categorised in the lower degree of processing).

Products’ images on circulars’ pages (examples in Appendix A) were analysed to identify the claims on the ‘front-of-pack’ (FoP) food labels. Illegible content was further investigated on manufacturers’ websites, or in situ at the supermarket stores. Identified claims were further classified as ‘Complementary Nutritional Information—CNI’ or ‘Additional Claims’. CNIs referred to ‘reduced amount or absence’ and ‘increased amount or presence’ of determined nutrients and energy value [39]. The terms identified and types of nutrient-content claims are presented in Table 1.

Claims on FoP labels that did not meet the criteria to be considered CNIs (i.e., ‘whole’, ‘organic’, ‘healthy’, ‘natural’ [40], were categorised as Additional Claims—AC. In Brazil, legislation determines that the presence of lactose and gluten in foods must be reported in the package labelling section containing the product’s ingredient list (usually on the back of the package) [41,42], to alert consumers who are allergic or intolerant. Therefore, the presence of such claims on FoP labels was characterised in the present study as AC.

Due to the large number of different CNIs and ACs identified on FoP labels, they were grouped into similar themes defined by two of the researchers and independently checked by a third researcher. Inconsistencies were discussed and resolved (Appendix A).

Information on the description of products was organised in a Microsoft Excel^®^ spreadsheet. Descriptive statistic was used to present data as absolute and relative frequencies, means, standard deviations (SD) or median, interquartile range (IQR) (depending on normality of distribution, assessed with Shapiro-Wilk test). Pearson chi-square test was used to compare the presence of increased and reduced nutrient-content CNIs and ACs in Unprocessed/Minimally Processed vs. Ultra-processed foods. Significance was established at *p* < 0.05. Stata version 13.0 (StataCorp, College Station, TX, USA) was used for data analysis. A post-hoc effect size analysis (w) of Pearson’s chi-square tests was conducted with G*power 3.1.9.2 considering an alpha of 0.05 [43].

Review by a Research Ethics Committee was not required for the study, as it did not involve human subjects.

## 3. Results

### 3.1. Health and Wellness Food Sections’ Characteristics and Degree of Processing of Advertised Food Items

Analysis of the 20 circulars obtained from the two supermarket chains (10 for each chain) led to the identification of 434 food items, with an average number of 21.7 (5.88 SD) foods per health and wellness section. Just over half (51.4%, *n* = 223) of the foods advertised were categorised as ultra-processed, followed by unprocessed/minimally Processed foods at 32.5% (*n* = 141), P foods at 8.7% (*n* = 38), and PCI at 7.4% (*n* = 32).

The three most frequently advertised ultra-processed foods were biscuits (21.1%, *n* = 47), processed cheese (10.8%, *n* = 24), and flavoured yoghurts (8.5%, *n* = 19), followed by vegetable-based beverages, granola, popcorn, ready-to-drink tea, and breads. The three most featured unprocessed/minimally processed food products were fruit juice (15%, *n* = 21), milk (13.6%, *n* = 19), and fish (14%, *n* = 10), followed by tapioca, fresh fruits, and ground coffee.

### 3.2. Complementary Nutritional Information—Reduced Amount/Absence

A total of 48 different CNIs were identified on the products’ FoP labels. From all the foods classified as unprocessed/minimally processed, and ultra-processed, 155 (43%) presented at least one CNI of reduced content or absence. Presence of this type of CNI was significantly higher (χ^2^ = 28.67, *p* < 0.001, effect size = 0.35) in the ultra-processed food group (77%, *n* = 120) when compared with the unprocessed/minimally processed group (23%, *n* = 35). The most frequent CNI of reduced content/absence was about sugar. Only foods from the ultra-processed group presented claims of ‘reduced content or absence of saturated fat’, ‘reduced energy content’, and ‘light’ (Figure 1).

### 3.3. Complementary Nutritional Information—Increased Content/Presence

From the foods classified as unprocessed/minimally processed and ultra-processed, 79 (22%) products presented at least one CNI of increased content or presence of a nutrient. This type of CNI was significantly more common (χ^2^ = 18.78, *p* < 0.001, effect size = 0.60) in the ultra-processed food group (82%, *n* = 65) when compared to the unprocessed/minimally processed group (18%, *n* = 14). The most frequent CNI was related to fibre content, which corresponded to more than 50% of the CNIs in both groups (Figure 2).

### 3.4. Additional Claims

A total of 136 claims classified as Additional Claims were identified on the FoP labels of foods advertised in the circulars. From the foods classified as unprocessed/minimally processed and ultra-processed, 269 (74%) presented at least one AC. The presence of ACs was not statistically different (χ^2^ = 0.29, *p* = 0.590) between unprocessed/minimally processed (38%, *n* = 102) and ultra-processed (62%, *n* = 167) groups. In the ultra-processed group, the most frequent ACs were ‘whole grain and fibre’ (21.7%, *n* = 36), ‘free-from’ or ‘low in lactose’ (15.1%, *n* = 25), ‘highlight on the presence of ingredients’ (10.1%; *n* = 17), and, ‘free-from gluten’ or ‘wheat-free’ (9.8%; *n* = 16) (Figure 3).

## 4. Discussion

This study identified and categorised food products advertised in health and wellness sections of supermarket circulars according to degree of processing, and identified claims on the front-of-pack labels of such products.

Firstly, we found that the most advertised group was of ultra-processed foods. The fact that the present research was limited to the health and wellness sections of supermarket circulars means that such materials may be negatively influencing consumers’ food purchases, contributing to the divergence of their diets from the recommendations of the Dietary Guidelines for the Brazilian Population [1]. In Brazilian supermarkets, as in many countries, circulars are made available to consumers online, in-store and also posted into their letter boxes with the intention to promote new products and present special offers [44]. Price affects consumers’ choices [45] and most purchases influenced by promotions are unhealthy, as reported in a systematic review [46]. Advertised products are further promoted in-store and positioned in strategic places such as end-of-aisle or islands located in places of great circulation, in order to influence shopping decisions [35]. In addition, circulars are reportedly used by consumers as planning tools [32,33] and can predict subsequent memory of the advertised product or brand [34]. One Brazilian study found that ultra-processed foods were three times more frequently advertised in supermarket circulars than unprocessed/minimally processed foods [31]. Another study analysed circulars from 12 Latin-American countries and reported that in six countries unhealthy foods were more advertised than healthy foods [24]. Our findings are supported by the aforementioned studies about supermarket circulars, but by focusing on the sections dedicated to the promotion of health and wellness-enhancing foods, we extend this stream of research to the phenomenon of increase in the population’s interest for healthy foods, reflected in selling strategies [14].

Another important finding was that ultra-processed foods presented significantly more CNIs than unprocessed/minimally processed foods. The most frequent CNIs in ultra-processed food packages were the increased content/presence of fibre and the reduction/absence of sugar. It is quite concerning from a public health point of view that ultra-processed foods are regarded as providers of fibre, because despite fibre content and absence of sugar, other undesirable ingredients such as hydrogenated vegetable fat, modified starch, food additives, artificial sweeteners or other formulations exclusively for industrial use which characterise these foods may also be present. The latter are responsible for increasing the palatability and shelf-life of ultra-processed foods [5,6]. Frequently, the presence of a CNI on labels of ultra-processed foods highlights a positive quality (usually a single nutrient), while negative aspects are not so visibly disclosed. In this sense, CNIs act as advertising strategies to promote sales instead of fulfilling their objective to provide more accessible information to the consumer [17,18]. Additionally, the term ‘premium’ is often used by the food manufacturing industry to refer to ultra-processed foods that, compared with ‘regular’ products, contain less ‘bad’ ingredients such as trans fats, sugar and salt, and more ‘good’ ingredients such as vitamins, minerals or whole foods such as fruits and nuts. While some of these modifications are positive, others may be harmful, as they will not make these products healthy, but mislead the consumer to think that they are [47]. The influence of nutrient claims on food products’ FoP over consumers’ perception has been established [48,49,50]. As Brazil does not have a nutrient profile system in place to evaluate the composition of foods, CNIs that highlight positive qualities of a food (e.g., high fibre content) can mislead the consumer when placed on products whose intake should be limited, such as ultra-processed foods. Brazilian researchers have already suggested that regulations on the use of nutrient claims in products bearing marketing strategies directed to children should be revised, so that only products with appropriate nutrient profiles should be allowed to display nutrient claims [51]. The same can be argued here, where foods advertised as healthy and wellness-enhancing are actually classified as ultra-processed due to their high degree of processing. These findings are cause for concern, as the consumption of ultra-processed foods is associated with higher body mass index and greater prevalence of both excess weight and obesity, as well as other non-communicable diseases [6,52].

Lastly, more than two-thirds of ultra-processed and unprocessed/minimally processed products advertised presented at least one additional claim (AC). Regarding ultra-processed foods, the most frequent ACs identified were related to whole grain, lactose and gluten content. All these claims may induce the consumer to perceive the products as healthier than they are. In spite of the information indeed being relevant to consumers with health issues, the ever-increasing interest by the industry to highlight the absence of lactose and gluten, for example, is not due to increased prevalence of celiac or lactose-intolerant individuals. What informs this strategy is the recent trend in the health and wellness sector to associate the free-from gluten and lactose products with a healthier lifestyle [53,54]. The study by Hartmann et al., (2018) evaluated the effect of free-from labels (including gluten and lactose) on consumers’ perception in four European countries, and discovered that consumers evaluated such products in a simplistic way and considered the ones labelled as free-from to be healthier [54]. In another recent study, individuals perceived products labelled as gluten-free as less energy-dense and less processed than similar ones containing gluten [55]. In respect to whole grain products, research revealed that consumers perceive whole grain foods as healthier, and more nutritionally balanced and natural than the refined ones [56] and although the innovative added-fibre to refined grain products may present a solution to increase fibre intakes [57], the mere addition of soluble or insoluble fibre to a product turns it into a ultra-processed food [6]. The ample use of health-related information on food labels and in the media can confuse the consumer [58] and trigger halo effects, in which a consumer thinks a product is healthier than it actually is [59,60] and lead to indulgence when eating them [61,62].

Educational level appears to influence the perception of free-from gluten and lactose products [63], as well as the concern with healthy eating and the practice of unhealthy weight loss strategies [21]. Up to today, no evidence of benefits for healthy individuals to avoid gluten has been obtained, but the idea is so common among certain social strata that the Brazilian Society for Food and Nutrition published a position paper stating that there is no evidence to support the fact that a gluten-free diet would be beneficial for a healthy individual [64]. The benefits attributed by healthy individuals to a lactose-free diet also remain unproven [54].

This study has some limitations. We were able to analyse circulars from only two supermarket chains in the city where the study was conducted, but this was due to the fact that only these two chains featured the health and wellness-enhancing section in their circulars. Additionally, there may be slight differences between circulars from distinct stores from the same chain, but as we collected circulars from only one store from each supermarket chain, we were not able to point those differences out. Regarding the higher presence of CNI of reduced content/absence and of increased content/presence of a nutrient in ultra-processed foods, the results were significant, but with small (0.35) and medium (0.60) effect sizes [65], respectively. Interpretation should be cautious, but amplifying data collection time would most likely not improve size effect as circulars were very similar to each other throughout the months. As we analysed claims present only on the FoP (and not on the other sides of packages) an underestimation of claims may have happened. However, it should be noticed that the FoP is also the first contact the consumer has with products.

To our knowledge, this is the first paper that evaluates the health and wellness-enhancing foods advertised in supermarket circulars. Our findings have important implications to promote healthy eating environments in Brazil. Results highlight to public policy-makers a scenario where food industry places CNIs and additional claims in ultra-processed foods’ FoP, which are further promoted by supermarkets as ‘healthy’ in circulars. Ideally, Brazilian legislation should be revised so that only information truly informing healthy choices to the consumer can be present in FoP labels of products. This is important because using successful advertising practices to promote healthy choices has the potential to enhance the health and well-being of consumers and reduce the expanding healthcare costs [66]. Attention should also be given by health professionals when guiding the population to healthier food choices within the supermarket environment, by disclosing this type of selling strategy.

Further research can be directed at exploring how consumers perceive the advertising of ultra-processed foods in health and wellness-enhancing sections of supermarket circulars and if, how do they shop in these sections. More studies aiming to identify the influence of health claims on FoP over consumer choices are needed, as a systematic review has demonstrated that although those claims are tested in healthy foods, they are mostly used in unhealthy foods [67]. Another potentially interesting area for future research is to investigate whether supermarket circulars in other countries use a similar strategy as in Brazil to promote ultra-processed foods as ‘healthy or wellness-enhancing’.

## 5. Conclusions

Results indicate that supermarket chains included in this study are promoting the sales of ultra-processed foods in the health and wellness-enhancing sections of their promotion circulars. This can lead consumers to inadvertently choose unhealthy foods when trying to adopt a healthy diet, which suggests a need for revision of Brazilian legislation regarding FoP labelling.

## Figures and Tables

**Figure 1 nutrients-12-02877-f001:**
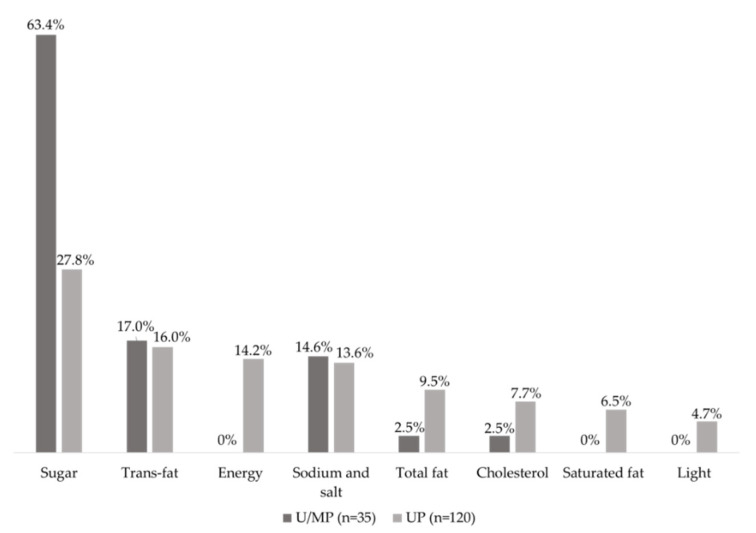
Distribution of Complementary Nutritional Information about ‘reduced content’ or ‘absence’ of different nutrients in the analysed products. Legend: U/MP = Unprocessed/Minimally Processed foods. UP = Ultra-processed foods.

**Figure 2 nutrients-12-02877-f002:**
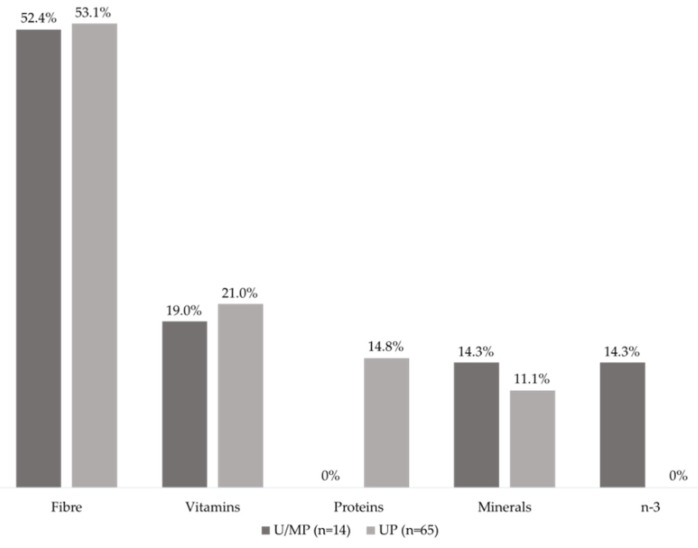
Distribution of Complementary Nutritional Information about ‘increased content’ or ‘presence’ of different nutrients in the analysed products. Legend: U/MP = Unprocessed/Minimally Processed foods. UP = Ultra-processed foods.

**Figure 3 nutrients-12-02877-f003:**
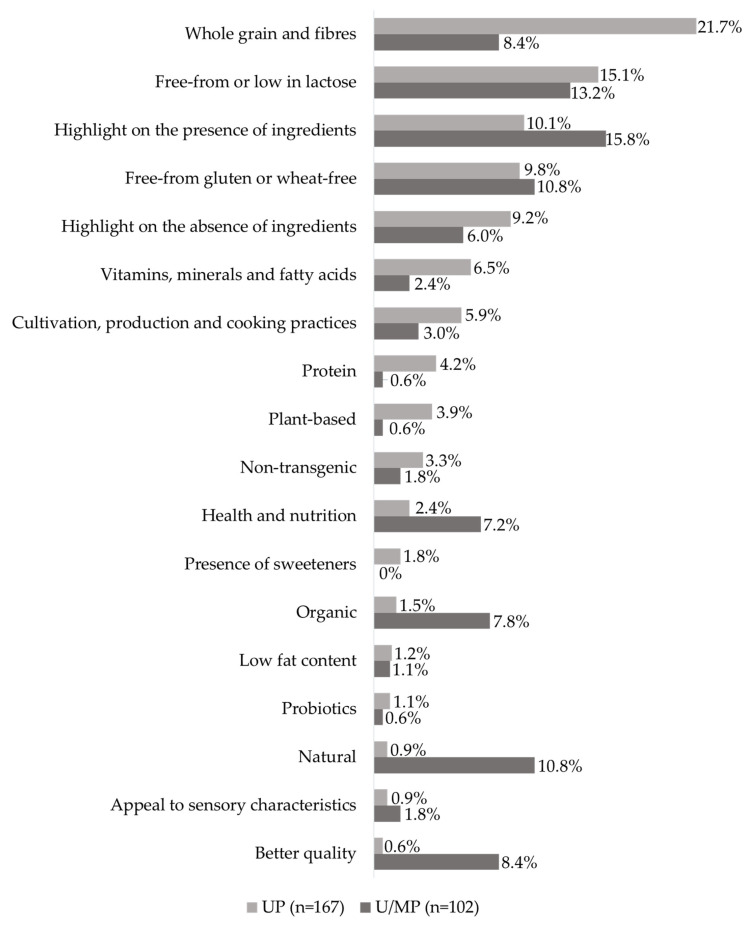
Distribution of Additional Claims in the analysed products, classified by themes (more information available in Appendix A—Appendix A). Legend: U/MP = Unprocessed/Minimally Processed foods. UP = Ultra-processed foods.

**Table 1 nutrients-12-02877-t001:** Types of claims, terms, and content/nutrients identified as regulated Complementary Nutritional Information—CNI.

Types of Claim	Terms Used	Content/Nutrient
‘Reduced amount or absence’	Low, reduced, light, free, very low, not added, zero, 0, 0%	Energy, total fat, saturated fat, trans-fat, cholesterol, sugar, sodium and salt
‘Increased amount or presence’	High, rich, with, contains, increased	*n*-3, *n*-6, and *n*-9 fatty acids, proteins, fibre, vitamins and minerals

Reference: elaborated by the authors based on Brazilian Legislation [39].

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
