# Peer review of "Supermarket Circulars Promoting the Sales of ‘Healthy’ Foods: Analysis Based on Degree of Processing"

_nutrients, 2020, doi:10.3390/nu12092877_

Round 1
Reviewer 1 Report
Title: Supermarket Circulars Promoting the Sales of ‘Healthy’ Foods: A Qualitative Analysis Focusing on Degree of Processing
This study assessed the presence of regulated and non-regulated front-of-pack food and nutrient claims in foods advertised in the health and wellness sections of circulars from two supermarket grocery chains in Brazil. The authors categorized the foods by degree of processing and compared the presence and types of claims between ultraprocessed (UP) and unprocessed/minimally processed (U/MP) foods. The authors found that UP foods were the most advertised group of foods, and that claims about the reduced amount/absence and increased amount/presence of certain nutrients were significantly higher in UP foods than U/MP foods. The authors’ suggested that these findings suggest that these claims may mislead consumers who may perceive the UP, less healthy foods to be more beneficial than they are.
This article focuses on a unique area of research that has important implications for consumer decision-making and promoting healthy dietary behaviors. However, this article should have a more substantial overview of the literature to set the context of dietary behaviors, FOP labeling policies, and influences/tactics used in supermarket advertising for readers. The article also requires editing for clarity in the use of English language and terms (which I will not focus on for this review) and greater clarity and elaboration of the methodology and rationale. Specific recommendations are outlined below.
Title
This research is not purely qualitative and should be revised accordingly. While the classification system may have required on qualitative characteristics, the use of statistical analyses are a quantitative research approach.
Introduction
Describe in greater detail the recommendations made in the Brazilian dietary guidelines in comparison to population diet patterns, especially with regards to nutrients of concern and level of processed food consumption (relevant to this research). How do processed foods contribute to the diet quality in Brazil?
Describe in greater detail the policies in Brazil/locally that guide or constrain the use of health and nutrient claims for international readers, particularly for CNI.
Add a description of the significance of Health and Wellness sections in supermarkets and advertisements for consumer decision-making. Do consumers perceive these products as more healthy because they are in this category; does it influence their purchase behaviors?
Materials and Methods
Please provide more information and rationale for the selection of the regional and national supermarket chains.
- How many stores are located in the city that they were identified from?
- Why was the city of Florianópolis selected?
- What volume of sales do these stores have and what variety of products are offered and most popular?
- Please describe the target audience/primary shoppers of these retailers (e.g., are they “value” stores that sell lower-price products or “high-end” stores that sell more expensive and/or higher quality foods?).
- Are there sociodemographic groups or characteristics of the shoppers that are of interest for health outcomes/dietary behaviors (e.g., income, education, transportation/physical accessibility)?
Describe the selection of circular advertisements for the two chains in greater detail.
- Did circulars cover all stores included in the target region or could individual stores have separate circulars?
- How many print vs online circulars were used for each store? If available, how many people are circulars distributed to or accessed by?
- Describe in detail what dates the circulars were extracted for and why these dates were selected for each chain. October 2018 – April 2019 is a seven month period, yet circulars were only extracted for 10 weeks.
- How frequently did each store release circulars?
Lines 99-100
It would be helpful for the reader to see an example of the circulars that were used in the analysis in a supplementary figure. Were there any nutrient claims made in the circular that were not on the FOP in the product image?
Results
Figures 1 & 2 - Include information so that the figures can stand alone for interpretation of the images. Include the number of products each processed food category.
Supplementary Table 1 - Some of the claims listed under themes in supplementary Table 1 do not seem to apply to the specific theme listed (examples below). Broad themes should be revised to encompass all claims listed within them.
- FREE FROM GLUTEN – a product is not necessarily gluten free just because it is wheat-free
- VEGAN – I suggest changing this theme to plant-based or some broader term to emphasize low/no animal and high plant product ingredients. The meaning of ‘100% veggie’ is not clear, and ‘vegetable-based ingredients’ are not clear enough to determine immediately if the product is truly vegan/free from any animal product derivatives.
- HIGHLIGHT ON THE ABSENCE OF INGREDIENTS – ‘100% blah blah blah’ does not refer to any real ingredients – please add a description of any supporting claims or elaborate on what this referred to.
- APPEAL TO SENSORY CHARACTERISTCS – what does ‘cereal letters’ mean?
Discussion
What are the limitations of the present research? How may have the limitations influenced the findings and interpretation of the present research?
Elaborate on the implications of these findings for future research, practice, and policy to promote healthy eating environments in Brazil.
Lines 175 – 178
Revise this sentence to clarify how the health and wellness section circulars may negatively influence food purchases or remove the sentence. This research did not look at the relationship between advertisement claims and consumer perceptions or purchase behaviors, so this claim should be further supported with research or removed.
Lines 181 – 184
How are UP foods ‘seen as providers of fibre’? The foods may actually provide fibre and could be a good source of fibre, even if they are UP.
Why is it more important for fruits and vegetables to be the main source of fibre? Why are fresh fruits and vegetables specified? Other forms of fruits and vegetables (e.g., frozen, canned, dried, juice) can play an important role in contributing to total fruit and vegetable intake.
Lines 202 – 209
The decision and rationale behind categorizing lactose- and gluten-free claims as ‘Additional’ instead of CNI claims needs to be reported in the methods section as it does not align with the regulated/CNI and unregulated/Additional categories defined previously.
Lines 217 – 223
Elaborate on prior research on consumer perceptions and purchase behaviors for other nutrient/ingredient claims covered in this research beyond just lactose and gluten.
Line 226
The use of the term ‘disclaiming’ seems to be inappropriate in this case (it means denying). Replacing the term with ‘disclosing’ appears to be more appropriate for what I understand as the intended meaning.
Author Response
This article focuses on a unique area of research that has important implications for consumer decision-making and promoting healthy dietary behaviors.
Thank you for your comment.
However, this article should have a more substantial overview of the literature to set the context of dietary behaviors, FOP labeling policies, and influences/tactics used in supermarket advertising for readers.
Thank you for the suggestion. A more comprehensive review of the pertinent literature on the suggested topics was included in the introduction section.
The article also requires editing for clarity in the use of English language and terms (which I will not focus on for this review) and greater clarity and elaboration of the methodology and rationale.
The manuscript was proof-read and methodology / rationale were revised.
Specific recommendations are outlined below.
Title
- This research is not purely qualitative and should be revised accordingly. While the classification system may have required on qualitative characteristics, the use of statistical analyses are a quantitative research approach.
Thank you for pointing this out. Title was revised and changed accordingly.
Introduction
- Describe in greater detail the recommendations made in the Brazilian dietary guidelines in comparison to population diet patterns, especially with regards to nutrients of concern and level of processed food consumption (relevant to this research). How do processed foods contribute to the diet quality in Brazil?
“The following paragraph was added (lines 34-41): Consumption of processed and ultra-processed foods has been growing exponentially in the Brazilian population and is considered a contributing factor for the increased prevalence of obesity and non-communicable diseases in the country [3,4]. Consequently, a new version of the Dietary Guidelines for the Brazilian Population was published, instructing individuals to limit the consumption of processed foods, avoid consumption of ultra-processed foods, and choose fresh and minimally processed foods as the core of their diets. According to the Guidelines, a healthy diet is based on the consumption of natural or minimally processed foods; and of dishes and meals containing such foods [1]”.
- Describe in greater detail the policies in Brazil/locally that guide or constrain the use of health and nutrient claims for international readers, particularly for CNI.
The following sentence was added (lines 96-103): “CNIs may refer to absolute or relative/comparative nutrient content of food products using terms as: ‘without’, ’no’, ’absence’, ’low content’, ’does not contain’; and ‘presence’, ’contains’, ’high content’, ’rich’, ’source of’. Regulation on parameters for the voluntary display of CNI on front-of-pack (FOP) labels of packaged products exist since 2012 [15], but claims are allowed without consideration for foods’ whole nutrient composition or degree of processing. Therefore, an ultra-processed food product containing high levels of sugar and/or sodium may display a nutrient-content claim of ‘low fat’ or ‘vitamin rich’ on its label”.
- Add a description of the significance of Health and Wellness sections in supermarkets and advertisements for consumer decision-making. Do consumers perceive these products as more healthy because they are in this category; does it influence their purchase behaviors?
The following sentence was added (lines 143-147): “The situation can mislead consumers and negatively impact their health, as supermarket circulars are reportedly used by consumers as planning tools [32, 33], and can predict subsequent memory of the advertised product or brand [34]. Foods advertised in circulars are also further promoted in-store and positioned in strategic places such as end-of-aisle or islands located in places of great circulation, in order to significantly influence shopping decisions [35].”
Materials and Methods
- Please provide more information and rationale for the selection of the regional and national supermarket chains:
How many stores are located in the city that they were identified from?
The following sentence was added (lines 167-169): “The national chain has a total of 29 stores in Brazil, of which six are located in Florianópolis; the regional chain has a total of 22 stores in the State and nine are located in Florianópolis.”
Why was the city of Florianópolis selected?
The following sentence was added (lines 159-160): “The capital was chosen out of convenience (near the university where the research team works), and also because it is the State’s second largest city...
What volume of sales do these stores have and what variety of products are offered and most popular?
Thank you for the suggestion. We add information about the gross sales of supermarket chains (lines 165-166), but were not able to obtain privileged information about variety of products or most popular products.
- Please describe the target audience/primary shoppers of these retailers (e.g., are they “value” stores that sell lower-price products or “high-end” stores that sell more expensive and/or higher quality foods?). Are there sociodemographic groups or characteristics of the shoppers that are of interest for health outcomes/dietary behaviors (e.g., income, education, transportation/physical accessibility)?
Thank you for the suggestion, but we do not have access to privileged information about target audience/primary shoppers. By experience, we know that they are neither ‘value’ stores nor ‘high-end’. We have added general information about the neighborhood in which the chosen stores are located (lines 170-171): “To collect the circulars, one store from each chain was conveniently chosen, both located in a residential middle-class neighbourhood near two university campuses.”
- Describe the selection of circular advertisements for the two chains in greater detail: Did circulars cover all stores included in the target region or could individual stores have separate circulars? How many print vs online circulars were used for each store? If available, how many people are circulars distributed to or accessed by? Describe in detail what dates the circulars were extracted for and why these dates were selected for each chain. October 2018 – April 2019 is a seven month period, yet circulars were only extracted for 10 weeks. How frequently did each store release circulars?
We added information in the second paragraph of the Methods section, except for the number of people who have access to the circulars, as we don't have access to this kind of information.
- Lines 99-100: It would be helpful for the reader to see an example of the circulars that were used in the analysis in a supplementary figure. Were there any nutrient claims made in the circular that were not on the FOP in the product image?
Images are now available as supplementary material. No nutrient claims were made in the circular, only on the FoP in products’ images.
Results
- Figures 1 & 2 - Include information so that the figures can stand alone for interpretation of the images. Include the number of products each processed food category.
The recommendations were accepted and text was amended accordingly.
- Supplementary Table 1 - Some of the claims listed under themes in supplementary Table 1 do not seem to apply to the specific theme listed (examples below). Broad themes should be revised to encompass all claims listed within them:
FREE FROM GLUTEN – a product is not necessarily gluten free just because it is wheat-free.
We changed the theme name to “Free-from gluten or wheat-free” in table S4, results section and, figure 3.
VEGAN – I suggest changing this theme to plant-based or some broader term to emphasize low/no animal and high plant product ingredients. The meaning of ‘100% veggie’ is not clear, and ‘vegetable-based ingredients’ are not clear enough to determine immediately if the product is truly vegan/free from any animal product derivatives.
We changed the theme name to “Plant-based” in table S4 and, figure 3. Regarding the expressions ‘100% veggie’ and ‘vegetable –based ingredients’, these were retrieved exactly as they appeared on the FoP. The reviewer is right to consider that they are not clear enough, this is exactly the problem with many of the Additional Claims identified in the products – they do not inform, only confuse. This is why we argue that they are actually marketing, not nutritional information.
HIGHLIGHT ON THE ABSENCE OF INGREDIENTS – ‘100% blah blah blah’ does not refer to any real ingredients – please add a description of any supporting claims or elaborate on what this referred to.
Thank you for the suggestion. The expression made reference to the product not having ingredients other than fruits. Again, this is the type of claim that does not inform and can actually confuse the consumer.
APPEAL TO SENSORY CHARACTERISTCS – what does ‘cereal letters’ mean?
Thank you for the contribution. We add to what this refers to in table S4: “Cereal letters (cereals with letter format - visual appeal)”.
Discussion
- What are the limitations of the present research? How may have the limitations influenced the findings and interpretation of the present research?
Thank you for the suggestion. We now describe the study’s limitations in the sixth paragraph.
- Elaborate on the implications of these findings for future research, practice, and policy to promote healthy eating environments in Brazil.
We added study implications for future research and practical recommendations in the seventh paragraph.
- Lines 175 – 178: Revise this sentence to clarify how the health and wellness section circulars may negatively influence food purchases or remove the sentence. This research did not look at the relationship between advertisement claims and consumer perceptions or purchase behaviors, so this claim should be further supported with research or removed.
We reformulated the whole paragraph.
- Lines 181 – 184: How are UP foods ‘seen as providers of fibre’? The foods may actually provide fibre and could be a good source of fibre, even if they are UP. Why is it more important for fruits and vegetables to be the main source of fibre? Why are fresh fruits and vegetables specified? Other forms of fruits and vegetables (e.g., frozen, canned, dried, juice) can play an important role in contributing to total fruit and vegetable intake.
We reformulated the third paragraph to address the reviewer’s concerns.
- Lines 202 – 209: The decision and rationale behind categorizing lactose- and gluten-free claims as ‘Additional’ instead of CNI claims needs to be reported in the methods section as it does not align with the regulated/CNI and unregulated/Additional categories defined previously.
We added the required information in the sixth paragraph of the methods section.
- Lines 217 – 223: Elaborate on prior research on consumer perceptions and purchase behaviors for other nutrient/ingredient claims covered in this research beyond just lactose and gluten.
Thank you for the suggestion. We added information in the fourth paragraph.
- Line 226: The use of the term ‘disclaiming’ seems to be inappropriate in this case (it means denying). Replacing the term with ‘disclosing’ appears to be more appropriate for what I understand as the intended meaning.
We thank you the correction, but after reviewing the discussion section this term was excluded.
Reviewer 2 Report
This manuscript investigates the quality of healthy foods advertised in the health and wellness-enhancing sections of supermarket circulars (flyers promoting food) from two Brazilian supermarket chains (one national and one regional). These foods were analysed according to degree of food processing and presence of claims on front-of-pack labels. The authors examine whether these ‘healthy’ foods promoted in the supermarket circulars are really healthy. The findings show that more than half of these ‘healthy’ foods are actually ultra-processed foods, which are not considered healthy. The main takeaway from the manuscript is that Brazilian supermarket circulars are found to be promoting the sale of ultra-processed foods in their health and wellness-enhancing sections, which can lead to a situation that potentially misleads consumers and negatively impacts their health.
This manuscript covers an interesting topic about a relatively ‘new trend’ in supermarket circulars (although I am wondering to what extent this is actually new?). The topic of healthy eating is important, because to prevent global obesity, there is not only a need for eating less unhealthy food, but even more importantly, there is a vital need to increase the consumption of healthy food among consumers. As such, it is necessary to upgrade the image of healthy food in the minds of consumers to further attempt to counter the obesity epidemic, which is also present in Brazil.
The Methodology and analysis of results have been well executed, however, the Introduction and Discussion parts of the research should be improved. I have made some suggestions to rewrite (mostly add to) these parts of the manuscript to better place this research in the broader context of healthy eating and healthy food promotion or advertising. In particular, the present manuscript could be much improved if it provided more reader confidence in it with a fuller awareness of previous research on supermarket circulars and healthy food promotion. The manuscript currently provides mainly practical value (which is important), however, it should be made more explicit how this study contributes to theory development. The contributions from this research should be stated more clearly, as well as to what extent this research replicates and extends previous research on food promotion in supermarket circulars. Please see below several suggestions and minor comments that may help to improve the manuscript.
Abstract
L14: Suggest to write “fourth-biggest market”.
L20: Suggest to write “ultra-processed foods” everywhere.
L18: Remove the extra spaces surrounding the slash for “increased/reduced”.
L20: Remove the extra spaces surrounding the slash for “presence/type”.
L22-23: Suggest to add the comparison foods group in: “Claims of increased fibre and reduced sugar content were significantly higher in labels of UP foods than in labels of U/MP foods”.
Introduction
In general, the Introduction should be extended more. Some definitions of essential concepts in the manuscript are lacking, as well as background on previous related research.
L30: Suggest to write “non-transmissible diseases”.
L38: Add a comma so that: “texture, and additional sensory properties”.
L41-42: You define ultra-processed (UP) foods, but you do not define unprocessed or minimally processed foods. Can you provide a definition of this food category as well, together with some examples? Similarly, since healthy food is the product category of focus in this research, can you define healthy food?
L43: Suggest to write “Despite” instead of “In spite of”.
L45-47: Can you elaborate further on this issue? As the importance of healthy eating is the starting point for this manuscript, some more in-depth interpretations of the problems surrounding the access to and adoption of healthy eating seem necessary.
L52: Are gluten and lactose-free foods considered as UP foods?
L54-57: Please add the ending apostrophe of the quotation.
L62: Suggest to use “health halo”. Can you elaborate further on the health halo effect?
L64: Suggest to add “Notably” or “Remarkably” to the beginning of this sentence, instead of using “even”.
L64-70: You refer to supermarket circulars here for the first time, but only in the following paragraph do you explain more about them. I would suggest to change the order of this part so that: “Notably, the ‘free-from’ consumption trend has influenced the content of supermarket circulars. Together with images of products’ front labels, these supermarket circulars present products’ prices, aiding the consumer to plan their shopping, but also significantly influencing their shopping decisions. National and regional supermarket chains have started to use whole sections of the circulars for the promotion of foods designated by them (possibly together with manufacturers), as health and wellness-enhancing”.
L67: What do you mean exactly with “wellness-enhancing”? Can you provide some food category examples for these sections of the circulars?
L70-72: One of the major concerns is that you do not explicitly argue to what extent your research differs from the previous research conducted on the content of supermarket circulars. How are the findings from your study different from the findings from [14-21], especially from [21]? You note the takeaway from these studies as: “This can mislead consumers and negatively impact their health”, but you argue for the same takeaway: “[…], leading to a situation that can mislead the consumer with negative health outcomes”. Can you state more explicitly to what extent your findings: (1) replicate previous research on supermarket circulars; and (2) corroborate or extend this stream of research, both in the Introduction as well as in the Discussion parts of the manuscript? Similarly, can you elaborate more on the findings of [14-21]?
Materials and Methods
L94-96: There are again some definitions missing of essential concepts used in the study. Can you provide the conceptualizations or operationalisations in the main text for the four food categories, possibly using a categorization table similar to Table 1?
L96-97: Can you elaborate on the decision flowchart?
L105: Add a comma and remove the extra spaces surrounding the slash so that: “Types of claims, terms, and content/nutrients…”.
L108: Suggest to write “Claims on FoP labels that did not meet the criteria…”.
L114: Suggest to write “Descriptive statistics were used…”, and add a comma between “means and standard deviations”.
L118: Suggest to write “StataCorp”.
Results
L133: Remove the extra spaces surrounding the slash for “amount/absence”.
L136: Can you provide effect sizes for the chi-square tests?
L138: Remove the extra spaces surrounding the slash for “amount/absence”.
L139: Add a comma between “reduced energy content, and light”.
L141: Concerning all the figures: I do not find the figures very convenient to interpret. Some minor adjustments might already improve the figures: placing the percentages above the bars instead of on the bars themselves, using dots instead of commas for the percentages, and providing a legend in the figures, either on the figure itself or in the figure caption.
Discussion
In your Discussion, you note three major findings: (1) the most advertised group in the health and wellness sections of supermarket circulars was of UP foods; (2) UP foods presented significantly more CNIs than U/MP foods, especially regarding the increase or presence of fibre and the reduction or absence of sugar; and (3) more than two-thirds of UP and U/MP products advertised presented at least one free-from additional claim (AC), about lactose or gluten content. Can you make these three findings more explicit, by stating that the results from the qualitative analyses point to three major findings, and using transition words in your Discussion (such as “first”, “second”, “third”, etc.)? In this way, you can make your contributions already more explicit.
L166-167: My major concern here is similar to my concern about the Introduction. Can you be more specific to what extent your findings (1) replicate previous research on supermarket circulars, and (2) extend this stream of research?
L180-181: Remove the extra spaces surrounding the slashes.
L191: Correct the typo in the sentence at “the overall nutritional status…”.
L208: Suggest to write “ever-increasing interest”.
L211-214: These findings imply a similar health halo effect as described in the Introduction, so I would suggest to elaborate on this.
L215: Suggest to write “energy-dense”.
L217: Suggest to write “the perception of…”.
L220: Suggest to write “strata of society” or “social strata”.
L229-231: The findings from your study are indeed especially relevant for health professionals and public policy-makers, however, you do not provide specific advice or recommendations. As this research currently has more practical value rather than providing theoretical contributions, I suggest to make these practical implications explicit. This is important because using successful advertising practices to promote healthy choices has the potential to enhance the health and well-being of consumers and reduce the expanding healthcare costs (e.g., see Bublitz, M.G.; Peracchio, L.A. Applying industry practices to promote healthy foods: An exploration of positive marketing outcomes. J. Bus. Res. 2015, 68, 2484–2493.).
L231-233: What do you mean exactly with “exploring how consumers perceive the advertising of ultra-processed health and wellness-enhancing foods”? Another potentially interesting area for future research is to investigate whether supermarket circulars in other countries use a similar strategy as in Brazil to promote ultra-processed foods as “healthy or wellness-enhancing”.
Conclusions
L235: Suggest to write “supermarket chains”.
L237-239: Here you do provide general practical advice for public policy, but you should elaborate on this earlier in the Discussion part [L229-231]. I also feel like this sentence is missing a part: “Ideally, Brazilian legislation should be revised so that only information truly informing healthy choices can be present in FoP labels of products in the healthy foods section of supermarket circulars”.
Other minor comments
Both the English and American-English spelling are used throughout the manuscript. Please use one spelling form consistently.
Author Response
Reviewer 2
This manuscript investigates the quality of healthy foods advertised in the health and wellness-enhancing sections of supermarket circulars (flyers promoting food) from two Brazilian supermarket chains (one national and one regional). These foods were analysed according to degree of food processing and presence of claims on front-of-pack labels. The authors examine whether these ‘healthy’ foods promoted in the supermarket circulars are really healthy. The findings show that more than half of these ‘healthy’ foods are actually ultra-processed foods, which are not considered healthy. The main takeaway from the manuscript is that Brazilian supermarket circulars are found to be promoting the sale of ultra-processed foods in their health and wellness-enhancing sections, which can lead to a situation that potentially misleads consumers and negatively impacts their health.
This manuscript covers an interesting topic about a relatively ‘new trend’ in supermarket circulars (although I am wondering to what extent this is actually new?). The topic of healthy eating is important, because to prevent global obesity, there is not only a need for eating less unhealthy food, but even more importantly, there is a vital need to increase the consumption of healthy food among consumers. As such, it is necessary to upgrade the image of healthy food in the minds of consumers to further attempt to counter the obesity epidemic, which is also present in Brazil.
The Methodology and analysis of results have been well executed, however, the Introduction and Discussion parts of the research should be improved. I have made some suggestions to rewrite (mostly add to) these parts of the manuscript to better place this research in the broader context of healthy eating and healthy food promotion or advertising. In particular, the present manuscript could be much improved if it provided more reader confidence in it with a fuller awareness of previous research on supermarket circulars and healthy food promotion. The manuscript currently provides mainly practical value (which is important), however, it should be made more explicit how this study contributes to theory development. The contributions from this research should be stated more clearly, as well as to what extent this research replicates and extends previous research on food promotion in supermarket circulars. Please see below several suggestions and minor comments that may help to improve the manuscript.
We thank the reviewer for the thorough analysis of the manuscript, and sought to address all the suggestions for improvement.
Abstract
- L14: Suggest to write “fourth-biggest market”.
Done.
- L20: Suggest to write “ultra-processed foods” everywhere.
Done.
- L18: Remove the extra spaces surrounding the slash for “increased/reduced”.
Done.
- L20: Remove the extra spaces surrounding the slash for “presence/type”.
Done.
- L22-23: Suggest to add the comparison foods group in: “Claims of increased fibre and reduced sugar content were significantly higher in labels of UP foods than in labels of U/MP foods”.
Done.
Introduction
- In general, the Introduction should be extended more. Some definitions of essential concepts in the manuscript are lacking, as well as background on previous related research.
Thank you for the suggestion. We believe that with the newly added paragraphs essential concepts are now reported as well as background on previous related research.
- L30: Suggest to write “non-transmissible diseases”.
Done.
- L38: Add a comma so that: “texture, and additional sensory properties”.
Done.
- L41-42: You define ultra-processed (UP) foods, but you do not define unprocessed or minimally processed foods. Can you provide a definition of this food category as well, together with some examples?
The following paragraph was added (lines 56-65): “Unprocessed (or natural) foods are edible parts of plants (seeds, fruits, leaves, stems, roots) or of animals (muscle, offal, eggs, milk), and also fungi, algae and water, after separation from nature [5]. Minimally processed foods, that together with unprocessed foods make up NOVA group 1 are unprocessed foods altered by industrial processes such as removal of inedible or unwanted parts, drying, crushing, grinding, fractionating, roasting, boiling, pasteurisation, refrigeration, freezing, placing in containers, vacuum packaging or nonalcoholic fermentation. None of these processes add salt, sugar, oils or fats, or other food substances to the original food. Their main aim is to extend the life of grains (cereals), legumes (pulses), vegetables, fruits, nuts, milk, meat and other foods, enabling their storage for longer use, and often to make their preparation easier or more diverse [6]”.We also added Supplementary material on this topic (Table S1).
- Similarly, since healthy food is the product category of focus in this research, can you define healthy food?
The following paragraph was added (lines 42-52): “Published in 2015, the Dietary Guidelines for the Brazilian Population established specific eating directives based on degree of food processing, employing a classification system later improved and published under the name of NOVA (a name, not an acronym) [5]. According to NOVA, industrial formulations containing little or no fresh ingredients and food additives to add colour, flavour, texture, and additional sensory properties to unprocessed foods and preparations containing them are classified as Ultra-processed (UP) foods [1,5]. The nature of the processes and ingredients used in their manufacture and their displacement of unprocessed or minimally processed foods and freshly prepared dishes and meals, make ultra-processed foods intrinsically unhealthy. On the other hand, dietary patterns based on dishes and meals made from a variety of unprocessed or minimally processed plant foods, prepared, seasoned and cooked with processed culinary ingredients and complemented with processed foods are the healthier ones [6].”
- L43: Suggest to write “Despite” instead of “In spite of”.
Done.
- L45-47: Can you elaborate further on this issue? As the importance of healthy eating is the starting point for this manuscript, some more in-depth interpretations of the problems surrounding the access to and adoption of healthy eating seem necessary.
The following text was added to the manuscript (lines 71-87):
“A definitive and universally accepted concept of healthy eating does not exist, but its association with better health and disease prevention is largely recognised [10]. During the second half of the last century, the increased availability and diversity of (un) healthy foods considerably modified the concept of what constitutes a healthy diet [11]. As the focus changed from a nutrient-based approach to a food-based one, food classification systems based on degree of processing were proposed [12]. In this context, the higher the processing degree to which a food has been submitted, the lower is the frequency in which it should be ingested as part of a healthy diet [5].”
“As ultra-processed foods tend to be energy-dense and low-cost, low energy cost could be one mechanism linking ultra-processed foods with high consumption and consequent negative health outcomes [13]. In Brazil however, the total cost of diets based on natural or minimally processed foods is still lower than the cost of diets based on ultra-processed foods. Relatively expensive perishable foods like some vegetables, fruits, and fish are and should be consumed with other natural or minimally processed foods that have lower prices, such as rice, beans, potatoes, cassava, and other staple traditional Brazilian foods. Calculations based on Brazilian household budget surveys show that diets based on fresh and minimally processed foods, and dishes and meals made with these foods and culinary ingredients, are cheaper than diets made of ultra-processed foods, as well as being healthier [1].”
- L52: Are gluten and lactose-free foods considered as UP foods?
No, but as explained in lines 100-101, nutrient content claims can appear on FOP labels regardless of the products’ degree of processing.
- L54-57: Please add the ending apostrophe of the quotation.
Done.
- L62: Suggest to use “health halo”. Can you elaborate further on the health halo effect?
The sentence was modified to include more info on the health halo effect (lines 114-119): “The highlight of positive characteristics as nutrient claims on front-of-pack labels can generate a ‘health halo’ effect, when consumers’ assessment of a single positive characteristic of the food affects their judgment about the quality of the food as a whole [20]. Health halos can be conferred by claims concerning just one nutrient, because consumers often make generalisations about the overall health of a product based on one piece of information found on labels [21]”
- L64: Suggest to add “Notably” or “Remarkably” to the beginning of this sentence, instead of using “even”.
Done.
- L64-70: You refer to supermarket circulars here for the first time, but only in the following paragraph do you explain more about them. I would suggest to change the order of this part so that: “Notably, the ‘free-from’ consumption trend has influenced the content of supermarket circulars. Together with images of products’ front labels, these supermarket circulars present products’ prices, aiding the consumer to plan their shopping, but also significantly influencing their shopping decisions. National and regional supermarket chains have started to use whole sections of the circulars for the promotion of foods designated by them (possibly together with manufacturers), as health and wellness-enhancing”.
Done.
- L67: What do you mean exactly with “wellness-enhancing”? Can you provide some food category examples for these sections of the circulars?
The following paragraph was added (lines 126-135): “‘Wellness’ refers to the positive, subjective state that is opposite to illness [22], an evolving process toward achieving one’s full potential [23]. Wellness is positive/affirming and holistic, and encompasses lifestyle, spiritual, and environment wellbeing domains; it also accounts for the physical, mental, and social domains implied in health, and thus health is dependent on sufficient wellness [23]. As consumers today are more health conscious than ever before, and the food and beverage industry is driven by consumer demand and popular trends, the health and wellness trend is increasingly prevalent. Health and wellness foods such as energy bars, gluten and dairy-free products, products containing organic/prebiotic ingredients, and fortified/functional foods are significantly more expensive than the regular offering, which makes them an expensive luxury in many emerging markets [14].”
- L70-72: One of the major concerns is that you do not explicitly argue to what extent your research differs from the previous research conducted on the content of supermarket circulars. How are the findings from your study different from the findings from [14-21], especially from [21]? You note the takeaway from these studies as: “This can mislead consumers and negatively impact their health”, but you argue for the same takeaway: “[…], leading to a situation that can mislead the consumer with negative health outcomes”. Can you state more explicitly to what extent your findings: (1) replicate previous research on supermarket circulars; and (2) corroborate or extend this stream of research, both in the Introduction as well as in the Discussion parts of the manuscript? Similarly, can you elaborate more on the findings of [14-21]?
The following paragraphs were added (lines 136-150):
“Research on the content of supermarket circulars found that most advertised foods are unhealthy and not conducive to the adoption of a diet in line with official recommendations [24–31]. Studies reported that circular offers did not contribute towards an environment that supports healthy eating behavior, but most only assessed the products advertised on the front pages and not in the entire circulars. Additionally, they were mostly conducted in European countries, North America and Australia. Only one study was identified reporting the analysis of Latin American (Brazilian) supermarkets’ entire circulars”.
“The situation can mislead consumers and negatively impact their health, as supermarket circulars are reportedly used by consumers as planning tools [32, 33], and can predict subsequent memory of the advertised product or brand [34]. Foods advertised in circulars are also further promoted in-store and positioned in strategic places such as end-of-aisle or islands located in places of great circulation, in order to significantly influence shopping decisions [35].”
“The present study seeks to extend this stream of research focusing on circulars not only advertising novel products or promotions, but dedicating sections specifically to the promotion of as health and wellness-enhancing foods.”
Materials and Methods
- L94-96: There are again some definitions missing of essential concepts used in the study. Can you provide the conceptualizations or operationalisations in the main text for the four food categories, possibly using a categorization table similar to Table 1?
Done. Please see Supplementary material (Table S1).
- L96-97: Can you elaborate on the decision flowchart?
The decision flowchart and more details on its application are provided in a previous publication - Botelho, A.M.; Camargo, A.M. de; Dean, M.; Fiates, G.M.R. Effect of a health reminder on consumers’ selection of ultra-processed foods in a supermarket. Food Qual. Prefer. 2019, 71, 431–437, doi:10.1016/j.foodqual.2018.08.017.
- L105: Add a comma and remove the extra spaces surrounding the slash so that: “Types of claims, terms, and content/nutrients…”.
Done.
- L108: Suggest to write “Claims on FoP labels that did not meet the criteria…”.
Done.
- L114: Suggest to write “Descriptive statistics were used…”, and add a comma between “means and standard deviations”.
Done.
- L118: Suggest to write “StataCorp”.
Done.
Results
- L133: Remove the extra spaces surrounding the slash for “amount/absence”.
Done.
- L136: Can you provide effect sizes for the chi-square tests?
We added information about effect sizes for the chi-square tests in the sections: material and methods (lines 215-216) “A post-hoc effect size analysis (w) of Pearson's chi-square tests was conducted with G*power 3.1.9.2 considering an alpha of 0.05 [43]”; and results section (lines 235, 246).
- L138: Remove the extra spaces surrounding the slash for “amount/absence”.
Done.
- L139: Add a comma between “reduced energy content, and light”.
Done.
- L141: Concerning all the figures: I do not find the figures very convenient to interpret. Some minor adjustments might already improve the figures: placing the percentages above the bars instead of on the bars themselves, using dots instead of commas for the percentages, and providing a legend in the figures, either on the figure itself or in the figure caption.
Thank you for the suggestion. The recommendations were applied to all the figures.
Discussion
- In your Discussion, you note three major findings: (1) the most advertised group in the health and wellness sections of supermarket circulars was of UP foods; (2) UP foods presented significantly more CNIs than U/MP foods, especially regarding the increase or presence of fibre and the reduction or absence of sugar; and (3) more than two-thirds of UP and U/MP products advertised presented at least one free-from additional claim (AC), about lactose or gluten content. Can you make these three findings more explicit, by stating that the results from the qualitative analyses point to three major findings, and using transition words in your Discussion (such as “first”, “second”, “third”, etc.)? In this way, you can make your contributions already more explicit.
Thank you for the suggestion. We revised the wording to meet the suggestions, as can be seen in the second, third and fourth paragraphs of the discussion section.
- L166-167: My major concern here is similar to my concern about the Introduction. Can you be more specific to what extent your findings (1) replicate previous research on supermarket circulars, and (2) extend this stream of research?
We reformulated all the second paragraph to meet the recommendations.
- L180-181: Remove the extra spaces surrounding the slashes.
Done.
- L191: Correct the typo in the sentence at “the overall nutritional status…”.
Done.
- L208: Suggest to write “ever-increasing interest”.
Done.
- L211-214: These findings imply a similar health halo effect as described in the Introduction, so I would suggest to elaborate on this.
Thank you for the suggestion. We added the information in the fourth paragraph.
- L215: Suggest to write “energy-dense”.
Done.
- L217: Suggest to write “the perception of…”.
Done.
- L220: Suggest to write “strata of society” or “social strata”.
Done.
- L229-231: The findings from your study are indeed especially relevant for health professionals and public policy-makers, however, you do not provide specific advice or recommendations. As this research currently has more practical value rather than providing theoretical contributions, I suggest to make these practical implications explicit. This is important because using successful advertising practices to promote healthy choices has the potential to enhance the health and well-being of consumers and reduce the expanding healthcare costs (e.g., see Bublitz, M.G.; Peracchio, L.A. Applying industry practices to promote healthy foods: An exploration of positive marketing outcomes. J. Bus. Res. 2015, 68, 2484–2493.).
Thank you for the suggestion. We added the study’s practical implications on the seventh paragraph.
- L231-233: What do you mean exactly with “exploring how consumers perceive the advertising of ultra-processed health and wellness-enhancing foods”? Another potentially interesting area for future research is to investigate whether supermarket circulars in other countries use a similar strategy as in Brazil to promote ultra-processed foods as “healthy or wellness-enhancing”.
We modified the last paragraph to include suggestions for further research. Hopefully the meaning is now clearer.
Conclusions
- L235: Suggest to write “supermarket chains”.
Done.
- L237-239: Here you do provide general practical advice for public policy, but you should elaborate on this earlier in the Discussion part [L229-231]. I also feel like this sentence is missing a part: “Ideally, Brazilian legislation should be revised so that only information truly informing healthy choices can be present in FoP labels of products in the healthy foods section of supermarket circulars”.
We revised our arguments in the discussion section. The sentence mentioned above was excluded from the text.
Other minor comments
- Both the English and American-English spelling are used throughout the manuscript. Please use one spelling form consistently.
The manuscript was revised and proof-read.
Round 2
Reviewer 1 Report
I have one minor comment:
Line 47 – Describe what makes ultra-processed foods “intrinsically” unhealthy. How is “The nature of the processes and ingredients used” related to diet quality? Specifically, what nutrients do these foods typically lack, and what nutrients of concern are they high in?
It would be helpful to discuss the reasons ultra-processed foods are popular (e.g., longer shelf life, lower cost, greater palatability) that lead to high consumption of these products.
If this is addressed, I do not have any other feedback and leave the rest to the editors. Thank you for the opportunity to review this manuscript.
Author Response
Reviewer 1
- Line 47 – Describe what makes ultra-processed foods “intrinsically” unhealthy. How is “The nature of the processes and ingredients used” related to diet quality? Specifically, what nutrients do these foods typically lack, and what nutrients of concern are they high in?
It would be helpful to discuss the reasons ultra-processed foods are popular (e.g., longer shelf life, lower cost, greater palatability) that lead to high consumption of these products.
Thank you for the suggestion.
The following sentence was added (lines 47-51): “In spite of this, as foods typically energy-dense, rich in sugar, fat, and salt, they are hyper-palatable and cheap, what contributes to their high consumption. Ultra-processed foods are poor in dietary fibre, protein, vitamins, and minerals; and additives contained in their formulation increase shelf-life without increasing their cost [1,5,6]”.